# Antimicrobial Efficacy of 7-Hydroxyflavone Derived from *Amycolatopsis* sp. HSN-02 and Its Biocontrol Potential on Cercospora Leaf Spot Disease in Tomato Plants

**DOI:** 10.3390/antibiotics12071175

**Published:** 2023-07-11

**Authors:** Halaswamy Hire Math, Raju Suresh Kumar, Bidhayak Chakraborty, Abdulrahman I. Almansour, Karthikeyan Perumal, Girish Babu Kantli, Sreenivasa Nayaka

**Affiliations:** 1P.G. Department of Studies in Botany, Karnatak University, Dharwad 580003, Karnataka, India; hmhalaswamy2063@gmail.com (H.H.M.); pallabchakraborty3@gmail.com (B.C.); 2Department of Chemistry, College of Science, King Saud University, P.O. Box 2455, Riyadh 11451, Saudi Arabia; sraju@ksu.edu.sa (R.S.K.); almansor@ksu.edu.sa (A.I.A.); 3Department of Chemistry and Biochemistry, The Ohio State University, 151 W. Woodruff Ave, Columbus, OH 43210, USA; perumal.11@osu.edu; 4Department of Life Sciences, PIAS, Parul University, Vadodara 391760, Gujarat, India; gibugiri@gmail.com

**Keywords:** mining soil, *Amycolatopsis* sp., HR-ESI-MS, NMR, 7-hydroxyflavone, biological activities

## Abstract

The actinomycete strain HSN-02 was isolated from the soil of a mining field in the Sandur region, Bellary, Karnataka, India. According to the morphological, cultural, physiological, and biochemical characteristics and the 16S rDNA sequence analysis, the strain HSN-02 was identified as *Amycolatopsis* sp. The antimicrobial activity strain HSN-02 presented stable and moderate inhibitory activity against human pathogens. In pot experiments in the greenhouse, the development of Cercospora leaf spot was markedly suppressed by treatment with the purified compound from the strain HSN-02, and the control efficacy was 45.04 ± 1.30% in *Septoria lycopersici*-infected tomato plants. A prominent compound was obtained from the fermentation broth of the strain HSN-02 using column chromatography and HPLC. The chemical structural analyses using UV, FTIR, HR-ESI-MS, and NMR confirmed that the compound produced by the strain HSN-02 is 7-hydroxyflavone. This investigation showed the role which the actinomycete strain can play in controlling leaf spots caused by *S. lycopersici* to reduce treatments with chemical fungicides.

## 1. Introduction

Bioactive metabolites are byproducts of various organisms’ primary and secondary metabolism (plants, animals, bacteria, and fungi). They frequently exhibit biological activity [1]. Secondary metabolites, unlike primary metabolites, serve no purpose in the cell’s life cycle and are used to identify specific groups of organisms. Over a million natural compounds have been described as a result of genetic techniques and high throughput screening (HTS) methods. Up to 50–60% of these chemicals are produced by plants (alkaloids, flavonoids, terpenoids, steroids, and carbohydrates), and bacteria produce 5%. Approximately 20–25% of all natural products reported have biological activity, and approximately 10% have been obtained from microbes. Actinomycetes produce 45% of all the biologically active compounds that have been found so far, fungi 38%, and unicellular bacteria 17% [2,3]. Until now, more than 140 genera of actinomycetes have been identified. Most of the known essential compounds come from a small number of these organisms [4,5]. Gram-positive bacteria with a high G + C content (over 55%) are classified as actinomycetes. This group includes prominent families, which account for nearly 75% of all known secondary metabolites, many of which are highly relevant to human health and the biotechnology sector [6]. In addition to their use as drugs (such as antifungals, antibacterial, immunostimulants, and tumor suppressors), herbicides, and growth promoters, these substances have a broad range of industrial and medical uses [7,8].

The fact that scientists are finding fewer new antimicrobial compounds and that bacteria are becoming resistant to antibiotics has made it vital for them to study unexplored habitats to find new actinobacterial isolates and bioactive compounds. *Streptomyces* is the actinomycete genus that produces the most secondary metabolites and has been the subject of the most research. However, members of the families *Pseudonocardiaceae* and *Micromonosporaceae* have also exhibited the capacity to produce a wide range of bioactive molecules. The genus *Amycolatopsis* belongs to the *Pseudonocardiaceae* family and produces several critical secondary metabolites, such as balhimycin, vancomycin, avoparcin, ristomycin, chelocardin, chloroeremomycin, and rifamycin [9]. So far, 70 different *Amycolatopsis* species have been identified and isolated from a wide range of environments, including soils, plants, ocean sediments, and clinical sources [10].

Members of the genus *Amycolatopsis* are nonacid-fast, nonmotile actinomycetes that are aerobic or facultative aerobic, Gram-positive, catalase-positive, and contain mesodiaminopimelic acid in their cell wall peptidoglycan [11]. Fatty acids lack mycolic acids and are abundant in *iso*- and *anteiso*-branched components [12]. The G + C content of genomic DNA ranges from 66 to 75 mol%, and the most common menaquinone type is MK9 (H4). Due to its ability to produce a diverse group of antibiotics and secondary metabolites, this genus has sparked intense scientific interest and focus. Therefore, more detailed research needs to be undertaken on this large group of rare actinobacteria to learn more about its hidden potential and diversity. Until now, several bioactive substances have been isolated from *Amycolatopsis* strains derived from various habitats, but no secondary metabolites have been reported from mining soil.

The severity of fungal diseases in *Solanum lycopersicum* (tomato) causes significant annual yield loss and decreased global-scale production. Among the different fungal diseases, the Septoria leaf spot (Cercospora leaf spot disease) caused by *Septoria lycopersici* (S. lycopersici) in tomato plants is the most devastating disease [13]. This disease develops at any stage of the development of the plants, and symptoms show the appearance of water-soaked circular spots on older leaves and become grey-colored tan with a dark brown margin. Later, the fungus can spread to different parts, such as young leaves, petioles, and stem [14]. Several widespread chemical fungicide applications have been practiced over recent decades to manage and control the Septoria leaf spot disease. However, these practices also accounted for adverse effects, such as increased chemical contamination of the soil, cost, environmental pollution, microbial imbalance, and increased health hazards. The biological control of plant fungal diseases using beneficial microbes to counteract the chemical applications is becoming the most promising alternative due to its cost-effective, environmentally safe features [15]. A wide range of beneficial microbes (bacteria, actinomycetes, and fungi) are documented for their biocontrol efficacy, production of bioactive metabolites, and ecofriendly nature.

*Amycolatopsis* sp. BCA-696 is isolated from the rhizosphere of chickpea and showed 60% of inhibition against the charcoal rot of sorghum caused by *Macrophomina phaseolina* [16]. The *Amycolatopsis* sp. strain 1119, which showed antagonistic activity against three fungal pathogens, Phytophthora capsici, Phytophthora drechsleri, and *Pythium ultimum,* and was positive for all four studied PGP traits, suppressed seed germination and plant growth [17]. In another study, *Amycolatopsis* sp. SND-1 isolated from *Cleome chelidonii* revealed the maximum inhibition against *Cercospora canescens* [18].

Plant sources are the most commonly studied isoflavonoids and flavonoids, while microorganisms as potential sources are relatively understudied. There have been considerable advancements in the understanding of isoflavonoids and flavonoids that have been derived from fungi and actinomycetes due to their low toxicity and considerable biological activity. Isoflavones, which are isolated from fungi and actinomycetes, can be categorized into three types, namely, simple isoflavones, isoflavone glycosides, and complex isoflavones. Most of these compounds exhibit antioxidant, antitumor, antimicrobial, and antigalactosidase activity [19,20,21,22]. There are reports that most of the isoflavones and flavones produced by actinomycetes are caused by the *Streptomyces* species; however, *Amycolatopsis* species and *Micromonospora aurantiaca* 110B have been reported to produce isoflavones and flavones [23]. The compound 7-hydroxyflavone was isolated from a methanolic extract of *Avicennia officinalis* L. and has shown good anticancer and antioxidant properties. *Withania somnifera* leaf extracts were analyzed using GC-MS and HPLC in recent studies and found several bioactive compounds. According to the report, the compounds 4-aminoheptane, n-undecanophenone, quercetin, and 7-hydroxyflavone have been studied and are used for a variety of biological and medical purposes [24,25].

This study aimed at the extraction and isolation of potent metabolites produced by *Amycolatopsis* sp. HSN-02 and its antimicrobial and biocontrol efficacy against Cercospora leaf-spot-causing *Septoria lycopersici*.

## 2. Results and Discussion

### 2.1. Isolation and Screening

In this work, 21 actinomycete strains were isolated from mining soil samples from the Sandur region, Karnataka, India. Actinomycetes isolated from such ecosystems found in mining sites can be used in biological applications and constitute a potentially important source of bioactive compounds [26]. The microbial strain of interest is seeded by a single streak in the center of the agar plate. After an incubation period depending upon the microbial strain, the plate is seeded with the microorganisms tested by single streak perpendicular to the central streak. After further incubation, the antimicrobial interactions are analyzed by observing the inhibition zone size. The primary screening of the isolates disclosed that the isolate HSN-02 showed the most potential to inhibit the growth of pathogens. The actinomycete isolates were then screened for antimicrobial activity against pathogens, and HSN-02 was determined to have the most potential among all isolates (Table 1). At the time the results were observed, the hydrogen ion concentration of the agar adjacent to the streak was determined in order to eliminate inhibition due to acidity alone. On the basis of this results obtained, we can differentiate which organism showed the good, moderate and weak activity. This indicated the organism’s capacity to produce solid antimicrobial compounds. Thereby, the isolate HSN-02 was used for further study. Similarly, Piyapat et al. [27] reported the isolation of three *Amycolatopsis* sp. from Thai soil samples. During the primary screening, the organisms showed significant antimicrobial activity against *Staphylococcus aureus* (*S. aureus*), *Bacillus subtilis* (*B. subtilis*), *Escherichia coli* (*E. coli*), *Kocuria rhizophila* (*K. rhizophila*), and *Pseudomonas aeruginosa* (*P. aeruginosa*). Many novel species of the genus *Amycolatopsis* were isolated from diverse environments, such as soil [28], natural caves [29], sea sediment [30], vegetable matter [31], plant roots [32], and clinical material [33].

### 2.2. Phylogenetic Analysis of Amycolatopsis sp. HSN-02

The phylogenetic analysis of *Amycolatopsis* sp. HSN-02 was accomplished using the sequencing of the 16S rRNA gene. The obtained sequence was 1371 bp in length and deposited in the NCBI with the accession number ON014420.1. During the BLAST analysis using EzBioCloud software, the sequence revealed a high degree of similarity (99.49%) with the relative sequence of *Amycolatopsis keratiniphila* DSM 44409 (LQMT01000206). The type strain sequences were retrieved, and a phylogenetic tree was constituted based on the neighbor-joining method (Figure 1). The phylogenetic tree revealed the ancestral relationship of *Amycolatopsis* sp. HSN-02 with *Amycolatopsis keratiniphila* DSM 44409, forming a single clad. The 16S rRNA gene sequencing is an essential criterion for identifying actinomycetes at their species level. This is because the 16S rRNA gene has hypervariable and conserved regions. The hypervariable regions provide species-specific signature sequences that enable the differentiation between various distinct microorganisms, such as bacteria, archaea, and microbial eukaryotes. In contrast, the conserved regions are used for universal primer binding sites [34]. This result followed the report of Okoro et al. [35], where the actinomycetes were isolated from soil samples from the Atacama Desert in Chile. The isolated actinomycete formed a subgroup with the genus *Amycolatopsis* based on the 16S rRNA gene sequencing.

### 2.3. Characterizations of Amycolatopsis sp. HSN-02

The *Amycolatopsis* sp. HSN-02 was subcultured on a starch casein agar (SCA) medium for morphological analysis. The colonies were flattened, dry, and powdery, with filiform margins. The organism was Gram-positive and showed whitish to grey-colored aerial (Figure 2A) and pale-yellow-colored substrate mycelia (Figure 2B), which often produced a yellow-colored soluble pigment (Table 2). The SEM analysis of the organism revealed branched fragmenting mycelia containing reniform spores (Figure 2C). According to the reports, temperature, pH, nitrogen and carbon sources, and other trace elements of the culture medium largely affect the production of diffusible pigments [36].

During physiological characterizations, the *Amycolatopsis* sp. HSN-02 exhibited optimum growth at 30 °C and moderate growth at 35 °C. The organism could not grow at 20 °C, although weak growth was reported at 25, 40, and 45 °C. A luxuriant growth was observed at pH 7.0, and weak growth was noticed at pH 8.0; however, no growth was reported at pH 5.0, 6.0, 9.0, and 10.0 (Table 2). The growth of *Amycolatopsis* sp. HSN-02 was significantly influenced by physiological parameters, like pH and temperature. Our study’s results indicate optimum growth at a pH of 7.0. Actinomycetes can withstand a wide range of temperatures. In our study, the optimal temperature for the growth and production of antimicrobial metabolites was determined to be 30 °C, confirming that the organism was a strict mesophile. Similar studies were undertaken by other researchers, who reported that the optimal pH and temperature were necessary for the growth and metabolism of actinomycetes [37,38]. The organism *Amycolatopsis* sp. HSN-02 revealed diverse biochemical characteristics through VITEK-2 analysis, and the result is documented in Table 3. The organism showed 16 positives and 30 negative results out of 46 tests. Enzyme activities, like BETA-XYLOSIDASE, L-lysine ARYLAMIDASE, leucine ARYLAMIDASE, etc., were positive for this organism. The organism could utilize D-MANNOSE as a carbon source, showing resistance for KANAMYCIN and POLYMIXIN_B.

The antibiotic sensitivity profiling of *Amycolatopsis* sp. HSN-02 was carried out using different antibiotic discs (Table 4). Out of eight antibiotics, the organism was sensitive to streptomycin, doxycycline–HCl, and vancomycin, and intermediate to clindamycin. However, the organism was susceptible to gentamycin. Antibiotics, like azithromycin, lincomycin, and chloramphenicol, did not affect the growth of *Amycolatopsis* sp. HSN-02. Here, we found that soil actinomycetes were significantly resistant to antibiotics. Interestingly, the isolate that showed the most resistance against antibiotics also showed antagonistic activity. Pal et al. [39] found that bacterial endophytes from the leaves of *Andrographis paniculata*, which include the actinobacterium *Micrococcus*, are resistant to antibiotics. Gousterova et al. [40] examined the biosynthetic capabilities of 26 thermophilic actinobacteria and their sensitivity to twelve antibiotics. Antibiotic-resistant strains are common in the environment irrespective of the human use of antibiotics. With the presence of plasmids, antibiotic resistance can transfer horizontally from one bacterium to another and between phylogenetically distant bacteria, contributing to the well-known problem of antibiotic resistance [41].

### 2.4. Extraction of Secondary Metabolites

The analysis helped to find out the cultural condition for the highest production of metabolites. The cultural conditions included the volume, pH, and incubation temperature of the medium, speed of rotation, and the inoculum volume. The availability of oxygen is the most crucial parameter in cell growth during the secondary metabolites production and fermentation process. The potential strain HSN-02 was cultured using the shake flask method using 3 L of SC liquid broth (pH-7.0, 30 °C, and 180 rpm) for 7 days with a preliminary inoculum of 30 mL of seed culture. Initially, the culture was routinely analyzed to determine good yield at alternative days (4th, 9th, and 14th days). Based on the yield quantity on the 21st day, the culture broth containing a blend of secondary metabolites was harvested and mixed with the organic solvent ethyl acetate. The extracted metabolites were concentrated, and 30.5 mg of brown-colored crude extract was obtained. Numerous studies have suggested that ethyl acetate is the best solvent for the extraction of bioactive compounds (antimicrobial and anticancer), particularly from the genus *Streptomyces* [42].

### 2.5. Purification of Ethyl Acetate Extract

The SCA medium was used for the extraction of bioactive compounds from *Amycolatopsis* sp. HSN-02. Bioactive compounds were then extracted from the fermented broth and purified. The crude HSN-02 extract was subjected to column chromatography using dichloromethane (DCM) and methanol (MeOH) (100:0, 80:20, 60:40, 40:60, 20:80, 0:100, *v*/*v*) to afford 15 fractions. Three of the fractions (six, seven, and eight) showed antibacterial activity against *S. aureus* and *E. coli*. All the active fractions were pooled and purified using reverse-phase high-performance liquid chromatography (RP-HPLC) at 200 to 400 nm. Based on the fractions collected, six fractions were selected for further analysis and categorized as F1–F6. Fraction (F5) exhibited good antibacterial activity against the pathogenic bacteria *S. aureus* and *E. coli*, whereas F1 exhibited poor antimicrobial activity, and some fractions (F2, F3, F4, and F6) failed to exhibit any activity against the pathogenic microorganisms. Hence, the fraction F5 was selected for structure elucidation. Numerous studies have suggested that ethyl acetate is the best solvent for the extraction of bioactive compounds (antimicrobial and anticancer), particularly from the genus *Streptomyces* [43,44]. A schematic representation of the detailed purification and fractionation of the compound is given in Figure 3.

### 2.6. Identification and Structure Elucidation

In the current study, the identification and structure elucidation of the purified fraction was performed using UV, FTIR, HR-ESI-MS, and NMR analyses. The compound was extracted as a pale-yellow amorphous powder with an unpleasant odor and was found to be completely soluble in dimethylsulfoxide (DMSO), methanol, and water. UV-Vis spectroscopy was used for the quantitative analysis. The UV absorption spectrum of the compound in MeOH was reported at 247 and 267 nm (Figure 4). The maximum absorbance peaks range between 215 and 270 nm, and the characteristics of the absorption peaks indicate a high polyene nature. In general, aromatic compounds are strong chromophores in the UV range; natural substances can also be determined using UV-Vis spectroscopy [45].

The FTIR spectrum of the purified compound produced numerous distinct bands at 3372, 2919, 1745, 1616, 1575, 1457, 1186, 1078, and 750 cm^−1^. The sharp peaks at 2115 and 1900 cm^−1^ indicated the existence of the C-C bonds of the alkynes; the peaks at 1621, 1572, and 1554 cm^−1^ were assigned to the alkenes; the 1507 cm^−1^ peak indicated the aromatics; and the 1492 and 1451 cm^−1^ peaks corresponded to the benzene ring (Figure 5). Similar reports about the FTIR analysis of the compound from *Amycolatopsis* sp. YIM 130642 [46] and *Actinomycetes* sp. [47] suggested the presence of various functional groups, such as alkanes, amines, phenols, carboxylic acids, and aromatic compounds. Through FTIR spectroscopy, this study illustrates the potential of the actinomycetes species in synthesizing antibiotics and further examines the antibacterial activity of diverse isolates. In the future, this study could be helpful in the fight against drug-resistant illnesses [48].

The mass analysis of the compound showed the prominent molecular ion [M + H]^+^ peak at *m*/z 239.0709, yielding the accurate mass of 238.0629 (Figure 6). The comparison of the retention time of the standard and the compound revealed a similar retention time at 3.42 and 3.43 min, respectively (Figure 7A,B). A comparative analysis of the antimicrobial activity of the standard drug vs. the isolated compound was performed against *S. aureus* and *E. coli*, and the results are shown in Appendix A. The ^1^H NMR spectra of the compound were recorded in a DMSO-*d*_6_ solution using tetramethylsilane (TMS) as an internal standard. The distinct proton signals that occurred in the spectrum pointed to a distinct chemical environment. The ^1^H-NMR data of the compound showed the following resonance: *δ*, ppm 6.88 (2H, m), 6.96 (1H, d, *J* = 2.4 Hz), 7.52 (3H, m, ring B), 7.85 (1H, d, *J* = 9.2 Hz), and 8.01 (2H, m, ring B) (Figure 8). These results suggest the presence of flavone moiety. The assignment of carbons in the molecular structure was confirmed using ^13^C NMR, which confirmed the presence of 15 carbon signals (Figure 9). The spectrum in the ^13^C NMR experiment matched quite well with the hypothesized molecular structure, and the proton signal shift was seen in the ^1^H NMR experiment (Table 5), and it was compared with the results of Suslow and Janeczko [49]. The red color (Figure 8 and Figure 9) indicates the NMR signals and green color (Figure 8) represents integration value of proton of the purified compound.

Therefore, based on the obtained data from NMR, HMBC, HSQC, and NOESY analyses, the structure of the compound was determined to be 7-hydroxy-2-phenylchromen-4-one or 7-hydroxyflavone with the chemical formula C_15_H_10_O_3_ (Figure 10A–D). The numbers 2–10 and 1′–6′ in the benzene ring corresponding to carbon atoms positioning in the elemental structure.

*7-hydroxyflavone*: pale-yellow colored amorphous powder; UV (MeOH): max nm 267 and 247; FTIR (KBr) νmax (cm^−^^1^) 3372, 2919, 1745, 1616, 1575, 1457, 1186, 1078, and 750; ^1^H, ^13^C NMR (400 MHz) data, see Table 5; DMSO-*d*_6_; HR-ESI-MS (+) = 239.0709 [M + H]^+^ (calcd. for C_15_H_10_O_3_^+^, 238.0629).

The 7-hydroxyflavone was previously reported for the first time from the methanolic extract of the leaves of *Avicennia officinalis* L. in the tropical mangrove ecosystem of the Andaman and Nicobar Islands [24]. The media blank was analyzed using HRMS (Appendix A), and there was no presence of 7-hydroxyflavone. When the media was inoculated with the * Amycolatopsis* sp. HSN-02, it implicated that the formation of the flavone compound occurred with the help of microbial synthesis. Similar results were obtained when media-changing experiments helped the formation of new compounds that were the microbial products of *Micromonospora aurantiaca* 110B [23]. The naturally occurring flavonoids are rarely observed; most of these are derived using a microbial synthetic approach. In the same scenario, a flavone compound named 7,3′-di-(γ,γ-dimethylallyloxy)-5-hydroxy-4′-methoxyflavone was isolated from the culture broth of the actinomycete *Streptomyces* sp. MA-12 [50].

### 2.7. Antimicrobial Activity and Minimum Inhibitory Concentration (MIC) Assay of 7-Hydroxyflavone

The antimicrobial activity of 7-hydroxyflavone was tested against two Gram-positive and one Gram-negative pathogens and one yeast strain. However, the compound could inhibit the growth of all the pathogens. The antimicrobial activity was assayed with 25–100 µL of a purified compound through the agar well diffusion method (Figure 11A–D). The maximum inhibition zone was recorded against *S. flexneri* (11 to 17.66 mm), and the minimum activity against *B. cereus* (6.33 to 15.66 mm) is shown in the bar diagram, respectively (Figure 11E). Compared to the Gram-positive pathogens, it was found that Gram-negative pathogens were more resistant. This could be attributed to the presence of lipopolysaccharide as a significant structural unit in the outer membrane of Gram-negative bacteria. In contrast, Gram-positive bacteria lack this protective barrier and are susceptible to metabolites. In previous studies, the compound amycophthalazinone A isolated from *Amycolatopsis* sp. YIM 130642 exhibited potent inhibitory activity against *S. aureus*, *S. typhi*, and *E. coli* [46]. Similarly, Alam and Jha [51] reported that the ethyl acetate extract of *Amycolatopsis* sp. ST-28 exhibited significant antimicrobial activity in *Staphylococcus aureus*. *Amycolatopsis balhimycina* and *Amycolatopsis orientalis* have been reported to possess antibacterial activity against methicillin-resistant *Staphylococcus aureus* strains [52].

The MIC values of the tested pathogens is shown in Table 6. The pathogen *S. aureus* showed the lowest MIC value of 1.6 µg/mL, and the highest MIC value of 6.25 µg/mL was recorded against the pathogen *S. flexneri*. It was evident from the MIC results that the Gram-positive pathogens were susceptible to lower concentrations, and the Gram-negative pathogens were susceptible to higher concentrations of the compound 7-hydroxyflavone. In similar studies, compounds isolated from *Amycolatopsis* sp. YIM 130642, namely, amycophthalazinone A, exhibited a potent inhibitory activity against *S. aureus*, *S. typhi*, and *C. albicans* with MIC values of 32, 32, and 64 µg/mL, respectively. In the same way, 7-O-methyl-5-O-α-L-rhamnopyranosylgenestein showed moderate inhibitory toward *S. aureus* and *E. coli* with an MIC value of 64 µg/mL [46].

### 2.8. Antifungal Assay

The purified compound of *Amycolatopsis* sp. HSN-02 displayed moderate antifungal activity against the fungi used in this study. The result of the antifungal activity of the purified compound is depicted in Figure 12A–D. The purified compound exhibited significant antifungal potential against *S. lycopersici* with 52.73 ± 1.04%, 73.68 ± 0.93%, and 90.61 ± 0.90% of mycelial inhibition at 25, 50, and 100 µg/mL concentrations of the compound, respectively, and the results are depicted in a bar diagram (Figure 13). The antifungal action of the bioactive metabolites can be induced by the effective action of the metabolites that cause the expansion and swelling of the mycelia of fungal pathogens. It is also facilitated by the ability of the metabolite, which counteracts the ergosterol production and synthesis of macromolecules. Previous reports suggested that 7-hydroxyflavone showed antifungal potential against *Cladosporium herbarum* (*C. herbarum*) and *Penicillium glabrum* (*P. glabrum*) pathogens [53].

### 2.9. Suppression of Cercospora Leaf Spot in Tomato Plants by Purified Compound

Leaf spots in tomato plants were measured after 4 weeks post inoculation of the *S. lycopersici* conidial suspension. The in vivo lengths of the developed spots were calculated in the control, pathogen-inoculated, and purified compound-treated tomato leaves (Figure 14A). The disease index was 0 for the control plants; in the pathogen-treated tomato plants, an 80.83 ± 1.63% of disease severity was noted. The tomato plants treated with a pathogen and the purified compound exhibited a significant reduction in disease severity of 45.04 ± 1.30%. The disease severity results in the tomato plants are graphically represented in Figure 14B.

Pretreatment with the purified compound from *Amycolatopsis* sp. HSN-02 suppressed leaf spot in the tomato plants caused by *Septoria lycopersici* in the soil without being phytotoxic to the tomatoes. Generally, soil-borne pathogens are controlled before planting using biological, chemical, and physical methods. In our present study, 7-hydroxyflavone indicated a fungicidal effect against *Septoria lycopersici*. These results suggest that it may be possible to sterilize the plant-pathogen-contaminated soil causes of tomato diseases, such as Cercospora leaf spot, by treating the soil with an inhibitory compound of the strain HSN-02 before planting. However, the strain HSN-02 has not yet been tested against other pathogenic fungi and bacteria that infect tomato plants, so further studies are required to investigate the potential of the purified compound against other tomato diseases. Our study suggests that the purified compound 7-hydroxyflavone derived from *Amycolatopsis* sp. HSN-02 can contribute to the development of new fungicides against the Cercospora leaf spot of tomato plants.

### 2.10. Toxicity against Eukaryotic Cells

The MTT assay results showed that the purified compound 7-hydroxyflavone do not caused any kind of toxicity against the HEK-293 cells by showing % cell viability greater than 90% after the incubation period of 24 h. The activity of the compound, which was evident by the distinct changes in shape, size, and other morphological changes are shown in the Figure 15A–G. Camptothecin was used as a standard or toxic control against the HEK-293 cells used for the current study. The cell viability was recorded as 99.89%, 99.62%, 98.63%, 97.03% and 93.52% at 12.5, 25, 50, 100, and 200 µg/mL concentrations of 7-hydroxyflavone (Figure 15H). In recent studies, the propolis extract was used against normal murine connective tissue cells L929, which exhibits less toxicity [54].

## 3. Materials and Methods

### 3.1. Pathogens Used in the Study

The pathogenic microbial strains, such as *Bacillus cereus* (*B. cereus*) (MTCC 11778), *Staphylococcus aureus* (*S. aureus*) (MTCC 6908), *Shigella flexneri* (*S. flexneri*) (MTCC 1457), and *Candida glabrata* (*C. glabrata*) (MTCC 3019), were procured from the Institute of Microbial Technology (IMTECH), Chandīgarh, India. The fungal phytopathogen was procured from the horticulture garden of the University of Agricultural Sciences, Dharwad, and subcultured on potato dextrose agar (PDA) media for further use. The chemicals and media used in the study were purchased from Hi-Media Laboratories Pvt Ltd. Thane, (West) 400604, Maharashtra, India and Sigma Aldrich Chemicals Private Limited, Bangalore, India.

### 3.2. Isolation and Screening

The soil samples were collected from the Sandur mining region during summer season (15°03′07.0″ N, 76°37′50.0″ E) at a depth of around ~15 cm using a sterile spatula and preserved at 4 °C in sterile polythene bags. The actinomycetes were obtained using the conventional serial dilution procedure on starch casein agar (SCA) (#M801, Hi-Media, Hi-Media Laboratories Pvt Ltd., Thane (West), 400604, Maharashtra, India) and incubated for 12 to 14 days at 30 ± 2 °C. The isolates were later purified and preserved on an SCA medium at 4 °C [55]. Primary screening of the actinomycete isolates was performed using cross-streak method against pathogens like *B. cereus*, *S. aureus*, *S. flexneri*, and *C. glabrata*. The actinomycete isolates were grown as straight lines on the middle of the separate petri plates on SCA medium (#M424, Hi-Media, Hi-Media Laboratories Pvt Ltd., Thane (West), 400604, Maharashtra, India) at 30 °C for 7 days. After incubation, the pathogens were streaked at a 90° angle to the actinomycete streaks and incubated for 24 h at 37 °C. Further analysis was carried out with the organism that showed the highest inhibition against pathogens.

### 3.3. Taxonomic Characterization

The molecular level identification of isolate HSN-02 was determined using sequencing of the 16S rRNA gene. The genomic DNA was extracted using HiPurA Streptomyces DNA purification kit (#MB527, Hi-Media, Hi-Media Laboratories Pvt Ltd., Thane (West), 400604, Maharashtra, India) following the manufacturer’s guidelines. The universal primers, 27F and 1492R, were utilized to perform PCR, which was programmed as follows: amplification of 35 cycles at 94 °C for 45 s, annealing at 55 °C for 60 s, and extension at 72 °C for 60 s. The cycle was repeated 30 times, and the final extension was carried out at 72 °C for 10 min. Later, PCR amplicons were validated using electrophoresis with a 1 kb of the reference DNA ladder. The DNA analyzer was used to sequence the desired PCR product, and the sequence was submitted to the gene bank through National Centre for Biotechnology Information website. BLAST analysis was performed to ascertain the phylogenetic neighbors of the isolate using the EzBioCloud platform (https://www.ezbiocloud.net, accessed on 6 April 2023). Finally, an evolutionary tree was constructed with similar type strains using MEGA7.0 software [56].

### 3.4. Morphological, Physiological, and Biochemical Characterizations

Morphological, physiological, and biochemical characterizations of *Amycolatopsis* sp. HSN-02 were determined according to the method of Bidhayak et al. [57]. For morphological characterizations, the colony characteristics, color of aerial and substrate mycelia, and pigmentation were recorded. To analyze the spore chain and spore surface morphology, the *Amycolatopsis* sp. HSN-02 was studied using a scanning electron microscope (SEM) (JSM-IT500, JEOL, Musashino, Akishima, Tokyo 196-8558, Japan). Physiological characterization was performed by growing the isolate at various temperatures ranging from 20 to 40 °C and pH ranging from 5.0 to 10.0. Biochemical properties were studied using the Vitek-2 BCL card test kit from BioMerieux (BioMerieux SA, 85 Voie Romaine, 69290 Craponne, France). Further, antibiotic sensitivity profiling was performed on several antibiotic discs, including chloramphenicol, gentamycin, azithromycin, lincomycin, clindamycin, doxycycline-HCl, streptomycin, and vancomycin on SCA medium [58].

### 3.5. Fermentation and Extraction

In the first stage, *Amycolatopsis* sp. HSN-02 was inoculated as seed culture in 50 mL of starch casein broth and grown for seven days at 30 °C. In the second stage, 3 L of starch casein broth in an Erlenmeyer flask was inoculated with 30 mL of seed culture and incubated for 21 days at the same temperature in a shaking incubator. After the complete growth of the organism, the biomass was separated using centrifugation at 5000 rpm for 15 min. To extract secondary metabolites, ethyl acetate solvent in a 1:1 (*v*/*v*) ratio was mixed with the supernatant, shaken intermittently, and left for 24 h for proper extraction of metabolites. The organic layer was separated and concentrated at 40 °C under low pressure using an IKA RV-8 rotary evaporator (IKA, Staufen, Germany).

### 3.6. Purification and Characterization

The concentrated ethyl acetate extract (EtoAc-Ex) was then subjected to column chromatography (35 × 1.0 cm) packed with silica gel (60–120 mesh size, #GRM7477, HiMedia). The fractions were eluted with an increasing polarity gradient solvent system of dichloromethane (DCM) and methanol (MeOH) (100:0, 80:20, 60:40, 40:60, 20:80, 0:100, *v*/*v*) to afford 15 fractions. Each fraction was examined for antibacterial activity against *S. aureus* and *E. coli*. The active fraction was pooled and purified again using reverse phase-high performance liquid chromatography (RP-HPLC). Elution was carried out with the solvent system of 70% isocratic methanol and water at a flow rate of 1 mL/min and at a wavelength range of 200 to 400 nm to yield a compound. The purified fraction was concentrated and checked for antibacterial activity against *S. aureus* and *E. coli*.

### 3.7. Identification of the Compound

The purified bioactive compound was analyzed using UV to inspect the wavelength absorbed by the compound using a Jasco V-670 UV-Vis Spectrophotometer (JASCO Corporation, 2967-5 Ishikawa-machi, Hachioji-shi, Tokyo 192-8537, Japan). To carry out the FTIR analysis, thin discs were made with KBr, and the infrared spectrum of the compound was recorded using a NICOLET 6700 (Thermo Fisher Scientific, Waltham, MA, USA). High-resolution electrospray ionization mass (HR-ESI-MS) spectrometry was performed using the Xevo G2-XS QTOF mass spectrometer (Waters Corporation, Milford, MA, USA). The instrument was equipped with an Accucore C-18 column (2.6 µm, 50 × 4.6 mm), and the temperature of the column was maintained at 40 °C. The mobile phase consisted of 0.1% formic acid and acetonitrile and was used to run the medium blank and standard (#H4530, Sigma-Aldrich Chemical Private Limited, Bangalore, India) and the purified compound from *Amycolatopsis* sp. HSN-02 to compare their retention times. The chromatographic eluent of the compound is in the range of 50 to 400 *m*/z. Nuclear magnetic resonance (NMR) spectroscopy was performed using an FT-NMR spectrometer (400 MHz, JNM-ECZ 400S, JEOL, 11 Dearborn Road, Peabody, MA 01960, USA), and ^1^H NMR and ^13^C NMR spectra were recorded. Trimethylsilane was used as an internal standard, and the chemical shifts of the compound were recorded in deuterated DMSO (DMSO-*d*_6_). Spin multiplets were reported as s (singlet), d (doublet), t (triplet), and q (quartet), and *J* was coupling constant in Hz.

### 3.8. Antimicrobial Activity and MIC Assay

Antimicrobial assay of the purified compound from *Amycolatopsis* sp. HSN-02 was performed against four pathogenic microorganisms using the agar well diffusion method. The working solution of the compound was prepared in DMSO (1 mg/mL), and antimicrobials, like streptomycin and amphotericin-B (1 mg/mL each), were used as positive controls. The pathogens were freshly cultured and 0.5 McFarland (1.5 × 10^8^ UFC/mL) concentrations were used for the antimicrobial assay. Mueller–Hinton (MH) agar (pH 7.4) plates were prepared and inoculated separately with 100 µL of test pathogens. Six mm wells were made, and the respective wells were filled with 25, 50, 75, and 100 µL of the compound. Sterile distilled water was employed as a negative control. The plates were incubated for 24 h at 37 °C. The antimicrobial assay was repeated thrice, and the mean zone of inhibition was calculated for each pathogen [56].

The minimum inhibitory concentration (MIC) of the pathogens was carried out according to the method of the Clinical and Laboratory Standards Institute (CLSI). The compound in a 100 µg/mL concentration was prepared in DMSO (99.8%), and streptomycin and amphotericin-B (1 mg/mL each) were used as positive controls. The microbial suspension was prepared (0.5 McFarland Standards), and 50 µL of each pathogen was dispensed in respective wells containing 200 µL of MH broth in 96 well plates. Hundred µL of the purified compound and the positive controls was pipetted out for pathogens and two-fold serially diluted up to Column 11. Column 12 alone was designated as sterility control, and incubation of the plates was performed at 37 °C for 24 h. After the incubation, 30 µL of resazurin (0.015%) was added to each well and incubated for 2 to 4 h for the observation of color change. Change in color from purple to pink indicated a positive response, and the lowest concentration at which color change was noted was the MIC value.

### 3.9. Antifungal Assay

An antifungal test was conducted against plant pathogenic fungi using a purified compound using the food poisoning technique [59]. To obtain the desired final concentrations (25, 50, and 100 µg/mL) compared to the control (without compound treatment), 250, 500, and 1000 µg of the purified compound was added to sterilized potato dextrose agar media (PDA) in Petri plates. Each Petri plate was poured with 10 mL of the purified compound-treated PDA. Following solidification of the PDA, 0.5 cm diameter discs from freshly grown tested fungi were placed in the middle of the plates. The control fungus was grown in an incubator at 28 °C until the plate was filled with its growth. There were three repetitions of all treatments. In order to determine the antifungal activity, the inhibition percentage of fungal hyphae was calculated as follows:% Inhibition = [Control − treatment/Control] × 100

### 3.10. Effect of Purified Compound on Cercospora Leaf Spot in Tomato Plants

Tomato seedlings at the first genuine leaf stage were inoculated with 10^6^ conidia/mL *Septoria lycopersici* using the root dip method (30 min) in the presence of 10 mL PD broth (5-day-old culture), then transferred to plastic pots (17 cm in diameter) containing commercial garden soil. All seedlings were maintained in a greenhouse at 28 °C with 14 h of light and 10 h of dark. Three weeks after inoculation, symptoms on tomato plants were scored on a scale of 0–3 (0 = no visible disease symptom on the fruit, 1 = slight infection, with small spots (≤1 mm), 2 = moderate infection, with medium spots (1–2 mm), and 3 = severe infection, with large spots (>2 mm)). The experiments were independently performed three times. The disease index (DI) was calculated using the following formula [60]:$$Disease\;Index\;(\%)\;=\;\frac{\sum \;scale\;\times \;number\;of\;spots\;per\;leaf}{Highest\;scale\;\times \;total\;number\;of\;spots\;per\;leaf}\;\times \;100$$

### 3.11. Toxicity against Eukaryotic Cells

The principle involved in the MTT assay is the reduction of tetrazolium salt into blue-colored formazan by the enzyme succinate dehydrogenase. The HEK-293 cell line was collected from NCCS, Pune. The cells were sub-cultured with 10% FBS and maintained at 37 °C for 24 h in a 5% CO_2_ atmosphere. Assays were conducted with medium without cells, control with cells but test compound, and standard with cells and 20 µM camptothecin. At a cell density of 20,000 cells/well, non-tumor cells were poured into 96 well flat bottom plates for 24 h. The cells were treated with 12.5–200 µg/mL concentrations of a purified compound for 24 h, followed by 0.5 mg/mL MTS reagent. After incubating, a gyratory shaker was used to gently shake, stir, and dissolve the MTT product in DMSO. The plates were encapsulated in aluminium foil and incubated for 3 h. Absorbance was recorded on an ELISA reader at 570 nm as a standard reference [61]. The percentage of inhibition was calculated, and the concentration of extract needed to inhibit cell growth by 50% (IC_50_) was determined by the dose-response curves for each cell line using the formula;
% of cell viability = [Absorbance of a sample at 570 nm /Absorbance of control at 570 nm] × 100.

### 3.12. Statistical Analysis

The results of at least three independent experiments are shown as means and standard deviations (SD).

## 4. Conclusions

The findings of the current study show that the mining soil actinomycete *Amycolatopsis* sp. HSN-02 produced a unique bioactive substance called 7-hydroxyflavone, which has broad spectrum antimicrobial and antifungal activity. Actinomycetes are found everywhere, and their morphological, physiological, biochemical, cultural, and molecular traits vary depending on the physicochemical parameters of their habitats. This study concludes that actinomycetes are everywhere. The untamed environment held the more significant potential for rare and novel actinobacterial communities, as well as for the extensive investigations required for isolations and creative and efficient taxonomic procedures that may result in the identification of novel genera and novel species of actinomycetes. It is necessary to look into previously undiscovered environments, particularly those near deep mines, to find new bioactive substances that might benefit humanity.

## Figures and Tables

**Figure 1 antibiotics-12-01175-f001:**
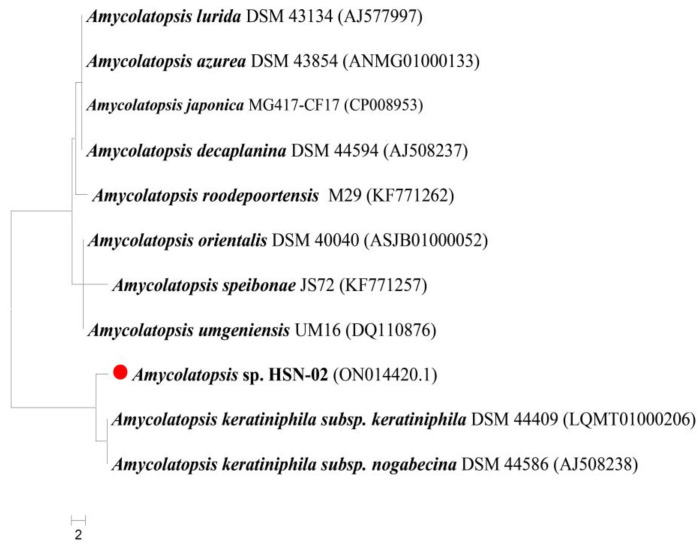
Phylogenetic relationship of *Amycolatopsis* sp. HSN-02 with neighboring *Amycolatopsis* species.

**Figure 2 antibiotics-12-01175-f002:**
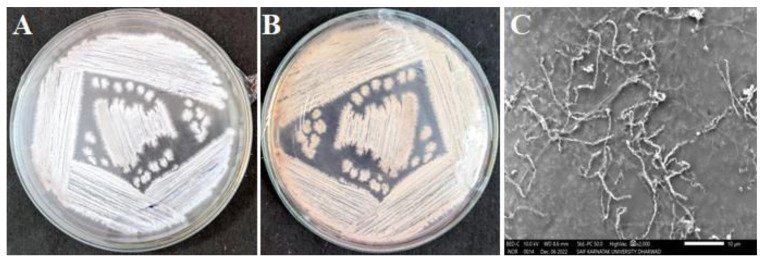
Morphological characterization of *Amycolatopsis* sp. HSN-02: (**A**) whitish-grey aerial mycelia, (**B**) pale-yellow substrate mycelia, and (**C**) SEM image depicting mycelia and spore chain morphology.

**Figure 3 antibiotics-12-01175-f003:**
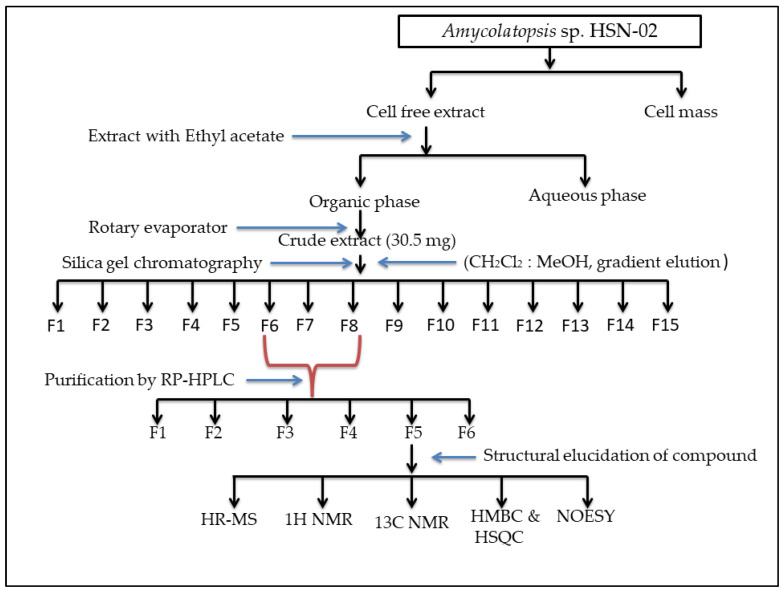
Schematic representation of purification and structural elucidation of the compound.

**Figure 4 antibiotics-12-01175-f004:**
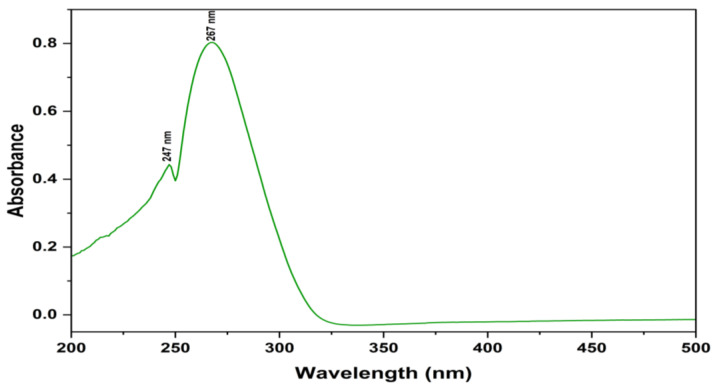
UV spectrum of 7-hydroxyflavone isolated from *Amycolatopsis* sp. HSN-02.

**Figure 5 antibiotics-12-01175-f005:**
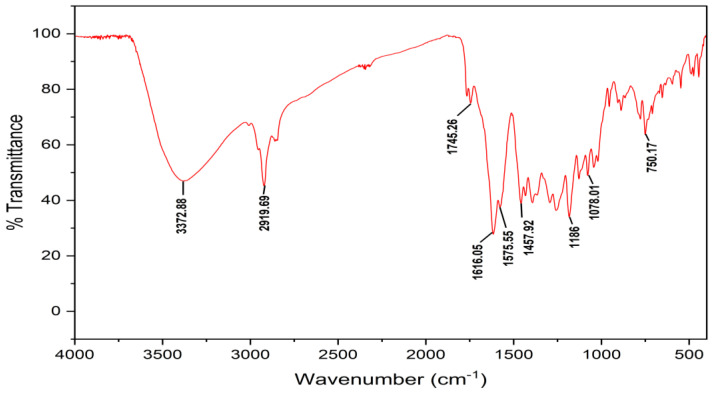
FTIR spectrum of 7-hydroxyflavone isolated from *Amycolatopsis* sp. HSN-02.

**Figure 6 antibiotics-12-01175-f006:**
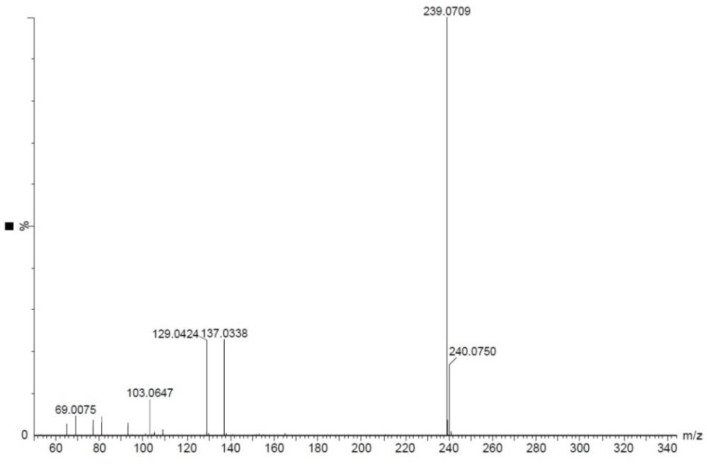
HR-ESI-MS spectrum of 7-hydroxyflavone showing a molecular peak at *m*/*z* 239.0709 [M + H]^+^.

**Figure 7 antibiotics-12-01175-f007:**
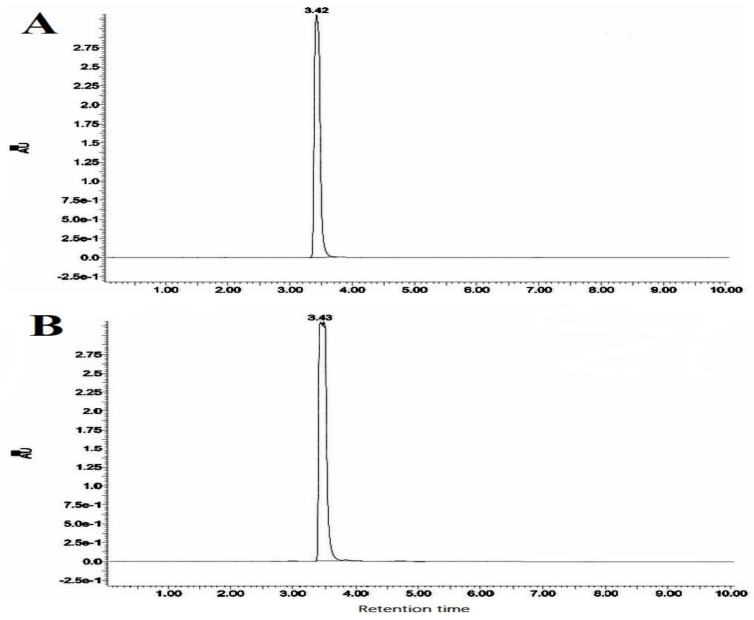
Elution profiles: (**A**) elution profile of the standard 7-hydroxyflavone (tR = 3.42 min) and (**B**) elution profile of 7-hydroxyflavone from *Amycolatopsis* sp. HSN-02 (tR = 3.43 min).

**Figure 8 antibiotics-12-01175-f008:**
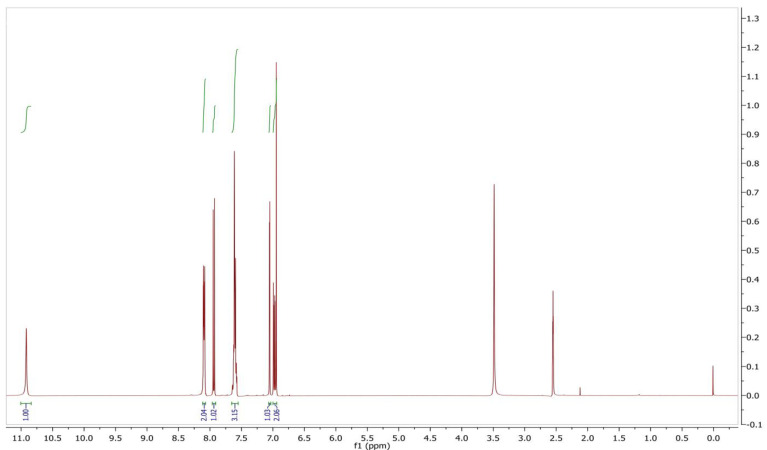
^1^H NMR spectrum of 7-hydroxyflavone.

**Figure 9 antibiotics-12-01175-f009:**
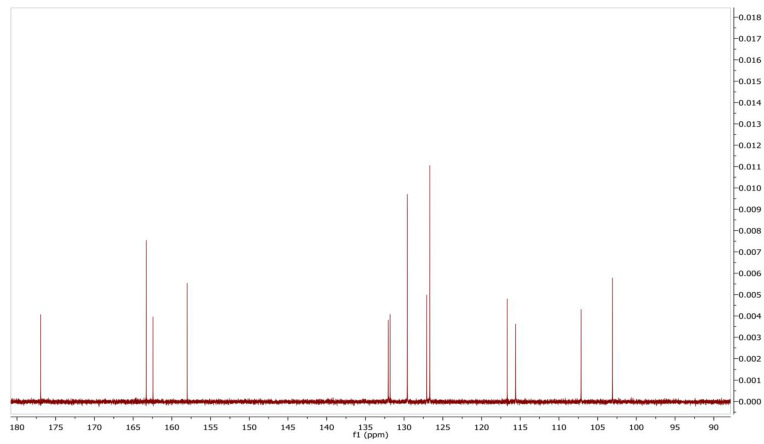
^13^C NMR spectrum of 7-hydroxyflavone.

**Figure 10 antibiotics-12-01175-f010:**
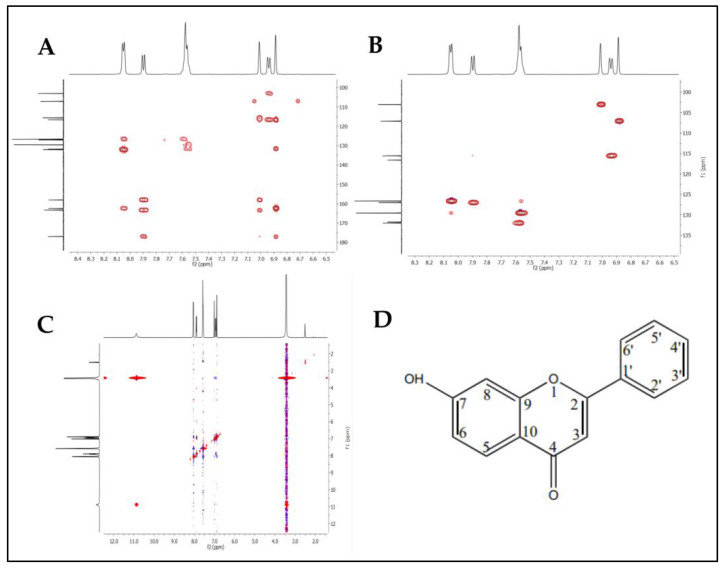
Structural prediction of 7-hydroxyflavone: (**A**) HMBC, (**B**) HSQC, (**C**) NOESY, and (**D**) chemical structure.

**Figure 11 antibiotics-12-01175-f011:**
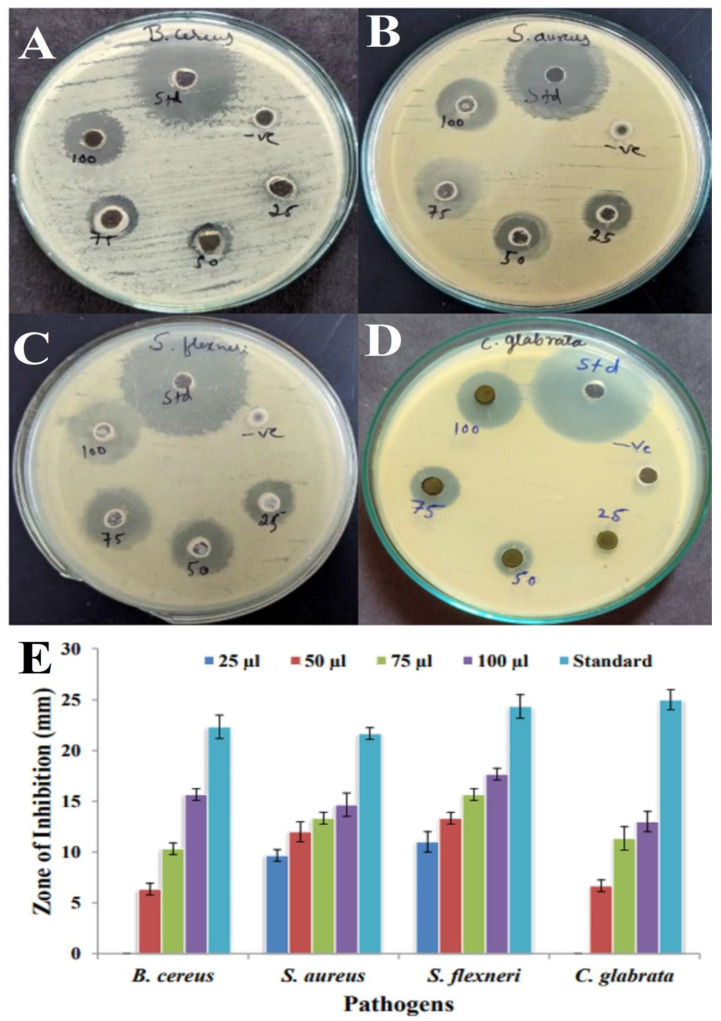
Antimicrobial activity of 7-hydroxyflavone using well diffusion method: (**A**) *B. cereus*, (**B**) *S. aureus*, (**C**) *S. flexneri*, (**D**) *C. glabrata*, and (**E**) bar graph showing zone of inhibition of pathogens at different concentrations of 7-hydroxyflavone.

**Figure 12 antibiotics-12-01175-f012:**
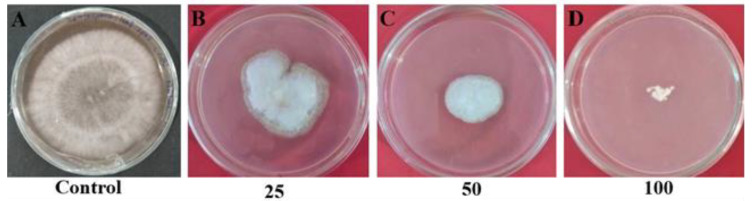
Antifungal activity of different concentrations of purified compound against *Septoria lycopersici*: (**A**) control, (**B**) 25 µg/mL, (**C**) 50 µg/mL, and (**D**) 100 µg/mL.

**Figure 13 antibiotics-12-01175-f013:**
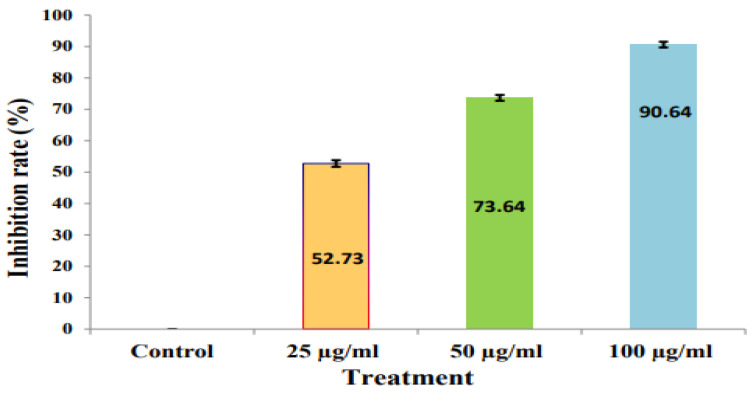
A histogram showing the inhibition rates of different concentrations of 7-hydroxyflavone on *Septoria lycopersici*.

**Figure 14 antibiotics-12-01175-f014:**
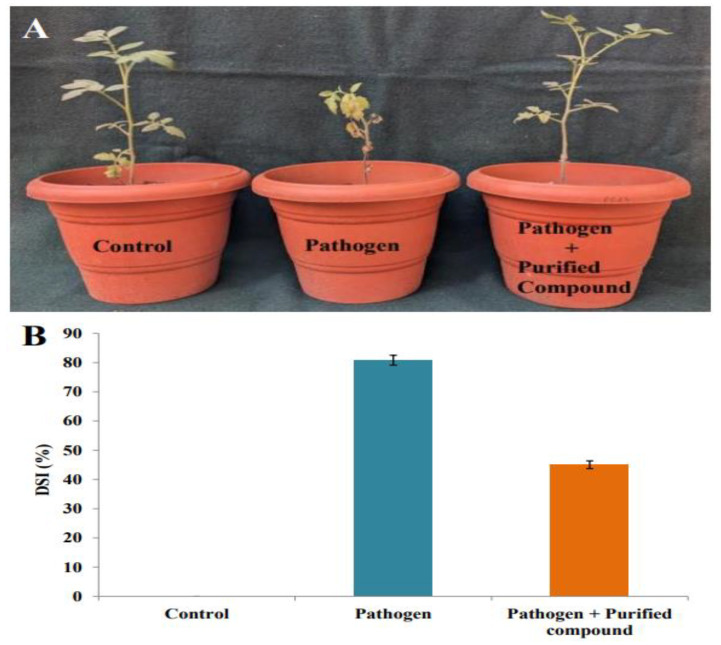
(**A**) The treatment of infected tomato plants with 7-hydroxyflavone against *Septoria lycopersici*. (**B**) Histogram showing effect of 7-hydroxyflavone on *Septoria lycopersici*-infected tomato plants.

**Figure 15 antibiotics-12-01175-f015:**
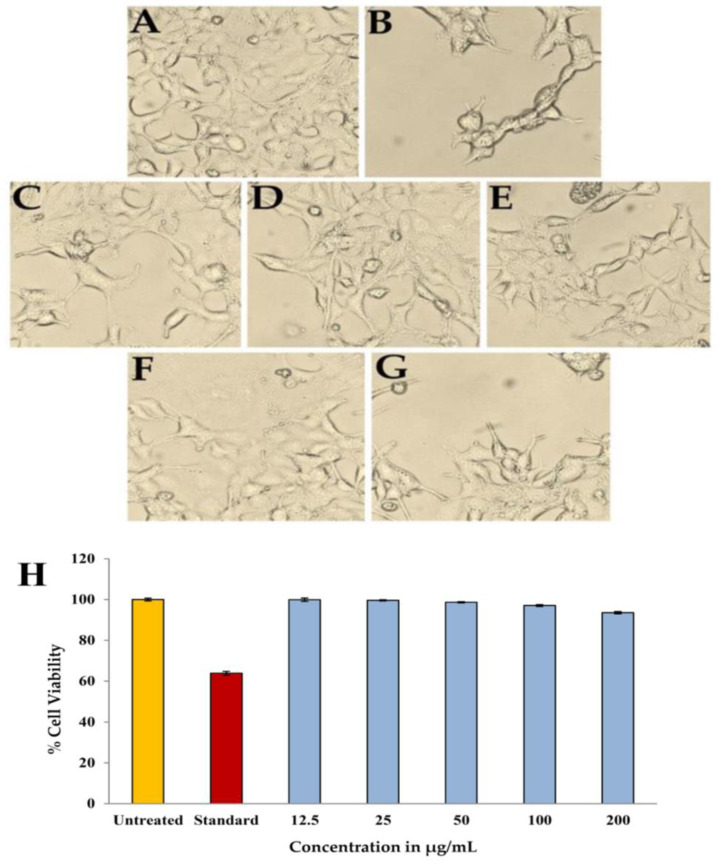
Morphological changes in HEK-293 cell line after treated with different concentrations of purified compound; (**A**) Untreated, (**B**) Standard (Campothecin), (**C**) 12.5 µg/mL, (**D**) 25 µg/mL, (**E**) 50 µg/mL, (**F**) 100 µg/mL, (**G**) 200 µg/mL and (**H**) Comparative % cell viability of HEK-293 cells treated with different concentrations of purified compound.

**Table 1 antibiotics-12-01175-t001:** Antimicrobial activity of actinomycete isolates against different test pathogens.

Actinomycetes Isolates	*B. cereus* (MTCC 11778)	*S. aureus* (MTCC 6908)	*S. flexneri* (MTCC 1457)	*C. glabrata* (MTCC 3019)
HM-01	+++	+++	+	+
HM-02	−	++	−	++
HM-03	−	+	+++	−
HM-04	−	+++	++	−
HM-05	++	−	−	+
HM-06	−	−	+	++
HM-07	++	+++	−	++
HSN-01	+++	++	++	+++
HSN-02	++	+++	+++	++
HSN-03	−	++	−	−
HSN-04	++	−	+	−
HSN-05	+++	+	+	++
HSN-06	+	++	+++	−
HS-01	+++	−	+	−
HS-02	−	+	−	++
HS-03	++	−	+	−
HS-04	+	+++	++	+++
HS-05	+++	−	−	+
HS-06	−	−	+++	−
HS-08	+	−	+++	−
HS-09	++	+	++	+

Note: +++ = good activity, ++ = moderate activity, + = weak activity, and − = no activity.

**Table 2 antibiotics-12-01175-t002:** Morphological and physiological characterizations of *Amycolatopsis* sp. HSN-02.

Morphological Characterizations				
Colony characters	Aerial mycelia		Substrate mycelia	
Color	Whitish to grey		Pale yellow	
Gram staining	Gram-positive			
Margin	Filiform			
Elevation	Flat			
Texture	Powdery and dry			
Pigmentation	Yellow colored pigment			
**Physiological Characterizations**				
Growth at different temperature		Growth in different pH		
20 °C	−	pH 5.0		−
25 °C	w	pH 6.0		−
30 °C	+++	pH 7.0		+++
35 °C	++	pH 8.0		w
40 °C	w	pH 9.0		−
45 °C	w	pH 10.0		−

Key: +++ = optimum growth, ++ = normal growth, w = weak growth, − = no growth.

**Table 3 antibiotics-12-01175-t003:** Biochemical characterisation of *Amycolatopsis* sp. HSN-02 using Vitek-2 Compact.

Tests	Results	Tests	Results
BETA-XYLOSIDASE	+	D-MANNITOL	−
L-Lysine-ARYLAMIDASE	+	D-MANNOSE	+
L-Aspartate ARYLAMIDASE	+	D-MELEZITOSE	−
Leucine ARYLAMIDASE	+	N-ACETYL-D-GLUCOSAMINE	−
Phenylalanine ARYLAMIDASE	+	PALATINOSE	−
L-Proline ARYLAMIDASE	−	L-RHAMNOSE	−
BETA-GALACTOSIDASE	−	BETA-GLUCOSIDASE	+
L-Pyrrolidonyl-ARYLAMIDASE	+	BETA-MANNOSIDASE	−
ALPHA-GALACTOSIDASE	−	PHOSPHORYL CHOLINE	−
Alanine ARYLAMIDASE	+	PYRUVATE	−
Tyrosine ARYLAMIDASE	+	ALPHA-GLUCOSIDASE	+
BETA-N-ACETYL-GLUCOSAMINIDASE	+	D-TAGATOSE	−
Ala-Phe-Pro ARYLAMIDASE	+	D-TREHALOSE	−
CYCLODEXTRIN	−	INULIN	−
D-GALACTOSE	−	D-GLUCOSE	−
GLYCOGEN	−	D-RIBOSE	−
myo-INOSITOL	−	PUTRESCINE assimilation	−
METHYL-A-D-GLUCOPYRANOSIDE acidification	−	GROWTH IN 6.5% NaCl	−
ELLMAN	−	KANAMYCIN RESISTANCE	+
METHYL-D-XYLOSIDE	−	OLEANDOMYCIN RESISTANCE	−
ALPHA-MANNOSIDASE	−	ESCULIN hydrolyse	+
MALTOTRIOSE	−	TETRAZOLIUM RED	−
Glycine ARYLAMIDASE	−	POLYMIXIN_B RESISTANCE	+

Key: + = positive result, − = negative result.

**Table 4 antibiotics-12-01175-t004:** Antibiotic sensitivity assay of *Amycolatopsis* sp. HSN-02.

Antibiotics	Zone of Inhibitions (mm)
Streptomycin (10 µg)	22, S
Clindamycin (5 µg)	16, I
Doxycycline–HCl (30 µg)	14, S
Vancomycin (10 µg)	11, S
Gentamycin (10 µg)	37, S
Azithromycin (15 µg)	R
Lincomycin (10 µg)	R
Chloromphenicol (30 µg)	R

Key: mm = millimeter, R = resistant, S = sensitive, and I = intermediate.

**Table 5 antibiotics-12-01175-t005:** ^1^H (400 Hz) and ^13^C NMR (400 MHz) data of 7-hydroxyflavone in DMSO-*d*_6_.

Comparative NMR Data				
	^a^* Published Data		^b^ Isolated Compound	
Position	*δ* _C_	*δ*_H_ (*J* in Hz)	*δ* _C_	*δ*_H_ (*J* in Hz)
2	162.8	-	162.4	-
3	106.6	6.86 (s)	107.14	6.96 (1H, d, *J* = 2.4)
4	176.4	-	176.9	-
5	126.5	7.86 (d, *J* = 8.7)	127.07	7.85 (1H, d, *J* = 9.2)
6	115.1	6.91 (dd, *J* = 2.3)	115.60	6.88 (1H, m)
7	161.9		163.3	-
8	102.6	6.97 (d, *J* =2.2)	103.05	6.88 (1H, m)
9	157.5	-	158	-
10	116.1	-	116.67	-
1′	131.3	-	131.8	-
2′	126.2	8.02 (d, *J* = 7.9)	126.68	8.01 (1H, m)
3′	129.1	7.52 (m)	129.59	7.52 (1H, m)
4′	131.6	7.52 (m)	132.1	7.52 (1H, m)
5′	129.1	7.52 (m)	129.59	7.52 (1H, m)
6′	126.2	8.02 (d, *J* = 7.9)	126.68	8.01 (1H, m)
7-OH		10.80 (s)	-	10.92

***** Published data from Suslow and Janeczko, 2012. ^a^ and ^b^ measured in DMSO-*d*_6__._

**Table 6 antibiotics-12-01175-t006:** MIC determination of 7-hydroxyflavone against different tested pathogens.

Pathogens	MIC (µg/mL)	Standards (µg/mL)
*B. cereus*	3.12	0.8
*S. aureus*	1.6	0.8
*S. flexneri*	6.25	1.6
*C. glabrata*	3.12	0.8

## Data Availability

Data available on request.

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
