# Peer review of "Antimicrobial Efficacy of 7-Hydroxyflavone Derived from Amycolatopsis sp. HSN-02 and Its Biocontrol Potential on Cercospora Leaf Spot Disease in Tomato Plants"

_antibiotics, 2023, doi:10.3390/antibiotics12071175_

Round 1

Reviewer 2 Report

Hello Dear colleague. Your manuscript is very interesting but some improvements are needed.

First your title is not clear. You should use either the word antimicrobial or antifungal, not both. I suggest: Antimicrobial efficacy of 7-Hydroxyflavone from Amycolatopsis sp. HSN-02 and its biocontrol potential on Cercospora leaf spot disease in tomato plant.

Next, regarding the abstract, at line 28, you wrote: “and results showed notable mycelia inhibition of % at 25….”. This sentence is not clear, which percentages? Kindly complete.

Let move to the introduction,

Lines 41-42: “Of these, major products (typically alkaloids, flavonoids, terpenoids, steroids, saccharides, etc.) are produced by plants, and least is produced by microorganisms [3]”. You said “least is produced by microorganisms” and at the next sentence you wrote “About 45% products are actinomycetes fermentation”. Do you mean that among all the produced Secondary metabolites, 45% are produced by actinomycetes? If so, kindly add at least 3 references which support this information.

Lines 46-47: kindly add a reference.

Lines 64-90: Kindly try to find how to make a good transition between the paragraph regarding Amycolatopsis and the one concerning fungal disease

I will start by material and methods before coming back to the results!

Kindly write the full name of each bacteria when you use it for the first time in any section ( B. cereus, S. aureus, S. flexneri…..)

At the section 3.2, kindly indicate the study period

Line 481: (#MB527, Hi-media), kinly add full information about the manufacturer

Line 513: which type of rota evaporator.? Kindly add information about the manufacturer

Line 547: 0.5 McFarland concentrations. Kindly add the approximative concentration of microorganisms in UFC/ml into brackets.

Line 551: if you speak about negative control here, kindly replace “standards” by positive controls in line 546

Line 556: Kindly add the concentration (%) of DMSO

Let come back to results and discussion

Line 98-99: Kindly add a reference

The section 2.4 is a kind a repetition of the material and method about extraction, kindly delete or add information about the results obtained.

Section 2.5 should be extended or better discussed

The other parts are well written and well designed.

Reviewer 3 Report

Reviver’s comments: Manuscript ID: antibiotics-2427169

The manuscript by Nayaka et al, describes the isolation and characterization of 7-hydroxyflavone from Amycolatopsis sp. HSN-02 and its antimicrobial, antifungal efficacy study. The manuscript is well written and results are presented well. The manuscript can be published after minor revision.

Comments:

1.     The biological importance of 7-hydroxyflavone is missing in the introduction part.

2.     In section 2.5; purification of ethyl acetate extract section the author has collected total 15 fractions. All fractions represent different compounds?

3.     Author also mentioned that the 3-fractions (F6-8) showed antibacterial activity. All three fractions have same compound or different compounds?

4.     1H NMR and 13 C NMR name legend 1 and 13 should be in superscript.

The quality of English is overall good but needs some minor formating.

Reviewer 4 Report

1. what do you mean by additionally characterized?? If it doesn't mean anything, pls remove "Additionally".

2. Line 23-26 and line 26-30 - make it a straightforward statement. Abstract should summarized the whole manuscript in a glance. If its dragging too much, repeating same thing again and agin, its not make interest for readers. Thus, Authors needs re-write the abstract again- straightforward, simple and attractive!

3. Line 52- I prefer not to use "AMONG OTHER THINGS". Pls change..

4. Line 91- metabolites

5. Line 92 - Capital Cercospora

6. Line 100-101 - Any primary data to support this statement? Such as preliminary antimicrobial activity of all the 21 isolates? etc

7. Line 107 - full form of the scientific names.

8. Line 113 - GeneBank acc#ON014420.1? check the number again.

9. Figure 1 - its better to have a more clearer figure.

10. Line 132 - SCA media full form

11. Line 198- what is the rational to select only these 8 antibiotics? I do not see these antibiotics represents different antibiotic classes.

12. line 216- little details about the media (solid vs broth) is needed

13. Line 221- a fractionation tree, solvent systems used for CC and HPLC gradient details needed to be included.

14. Line 226 - how does the authors determine the fractions were active? against control drug? Details needed and include into the text.

15. Line 230 - at this point, how do you know there is one active compound? and HPLC analysis of the active fractions?

16. line 234- ascending order  

17- Line 238-239 - not clear and not connecting to the para..

18 Line 242 - in FTIR what about OH and C=O stretching?

19. Line 263 -275- use correct notations

Line 316 - follow tabulating NMR data from published MDPI paper. ii) since its a already known include the comparison NMR chemical shift, coupling constant data table for your compound vs published data

Currently, for antimicrobial activity studies disk diffusion data is not accepted for pure compounds. Microbroth dilution is the standard method for reporting antimicrobial activity. Because, in the Figure 10- A-B-C-D the pathogenic bacterial inoculation was not even.  Especially A and D has a huge difference.

Since the Authors has pure cpds (used for elusion profile Fig 6), its better to study the head-to-head comparison if standard drug vs isolated cpd antimicrobial activity. 

Line 333- there are many publications mentioned that flavone type compounds are produced from the microbial cultures as a result of ingredients that are used for media preparation. Especially, if plant based ingredients such as soy, starch. To answer this concern the best way is analyzing the media blank by targeted analysis for the flavone compound through HRMS. I recommend  Authors analyze the media blank and include those data and claim that the isolated bioactive compound is a true microbial metabolite. 

In summary, I am not satisfied with the presented work as there are more space to improve. I recommend the Authors to go through the manuscript again, improve and resubmit.  Attached highlighted manuscript. 

needs extensive English editing 

Round 2

Reviewer 1 Report

The authors made the changes and corrections requested by this reviewer.

Therefore I propose that the article be published in this version.

Reviewer 4 Report

Authors have do attempted full fill all the comments and suggestions I made during the review process. I would like to include the NMR solvents that used to acquire the NMR data to the table in both cited reference and authors experiments. 
